# Expanding Our Knowledge of *DICER1* Gene Alterations and Their Role in Thyroid Diseases

**DOI:** 10.3390/cancers16020347

**Published:** 2024-01-13

**Authors:** Maria Cristina Riascos, Anh Huynh, William C. Faquin, Vania Nosé

**Affiliations:** 1Department of Pathology, Massachusetts General Hospital, Harvard Medical School, 55 Fruit Street, Boston, MA 02114, USA; mriascosmorillo@mgb.org (M.C.R.);; 2Mass General Brigham, Massachusetts General Hospital, Harvard Medical School, 75 Francis Street, Boston, MA 02115, USA

**Keywords:** *DICER1*, thyroid disease, familial tumor syndrome, thyroid follicular nodular disease, differentiated thyroid carcinoma, thyroblastoma, poorly differentiated thyroid carcinoma

## Abstract

**Simple Summary:**

Mutations in *DICER1*, a gene involved in RNA interference, have been associated with a wide range of multi-organ neoplastic and non-neoplastic conditions. Historically known for its association with pleuropulmonary blastoma, DICER1 syndrome has received more attention due to the association with newly discovered diseases and tumors. Recent studies evaluating *DICER1* mutations and DICER1-driven thyroid disease in both pediatric and adult thyroid nodules revealed thyroid disease as the most common manifestation of *DICER1* mutations. Thyroid follicular nodular disease and differentiated thyroid carcinomas in infancy are highly specific manifestations of germline *DICER1* mutation or DICER1 syndrome. Furthermore, poorly differentiated thyroid carcinoma and thyroblastoma should raise the concern for somatic *DICER1* mutations. Recognizing these manifestations should prompt clinicians to expedite genetic evaluation for this neoplastic syndrome and classify these patients as high risk for additional multi-organ malignancies.

**Abstract:**

Mutations in *DICER1*, a gene involved in RNA interference, have been associated with a wide range of multi-organ neoplastic and non-neoplastic conditions. Historically known for its association with pleuropulmonary blastoma, DICER1 syndrome has received more attention due to the association with newly discovered diseases and tumors. Recent studies evaluating *DICER1* mutations and DICER1-driven thyroid disease in both pediatric and adult thyroid nodules revealed thyroid disease as the most common manifestation of *DICER1* mutations. This study undertakes a comprehensive investigation into *DICER1* mutations, focusing on their role in thyroid diseases. Specific attention was given to thyroid follicular nodular disease and differentiated thyroid carcinomas in infancy as highly indicative of germline *DICER1* mutation or DICER1 syndrome. Additionally, poorly differentiated thyroid carcinoma and thyroblastoma were identified as potential indicators of somatic *DICER1* mutations. Recognizing these manifestations should prompt clinicians to expedite genetic evaluation for this neoplastic syndrome and classify these patients as high risk for additional multi-organ malignancies. This study comprehensively synthesizes the current knowledge surrounding this genetically associated entity, providing intricate details on histologic findings to facilitate its diagnosis.

## 1. Introduction

Combinations of multiple endocrine neoplasms occurring in families have been known long before the discovery of their common underlying genetic alterations. Since then, the number and knowledge of these syndromes have rapidly increased with the advent of widespread molecular diagnostics, which currently dominates clinical-oncologic practice.

The first endocrine familial syndromes were described in 1903 and 1953 when multiple endocrine neoplasia (MEN) syndromes 1 and 2, respectively, were encountered in autopsy examinations [1]. The familial correlation was described even before the existence of genetic testing. Now into the 21st century, a new set of endocrine familial syndromes has been described, including MEN 4, MEN 5, and MAFA-related insulinomatosis [2,3,4,5].

Following the clinicopathologic description of pleuropulmonary blastoma (PPB), a high-grade solid neoplasm of infancy, a collection of similar cases with a familial-patterned distribution was reported that also included other extrapulmonary tumors [6]. Due to the rapidly increasing use of DNA sequencing in diagnostic pathology, cytopathology, molecular pathology, and clinical genetics, a variant of *DICER1*, a gene involved in RNA interference (iRNA) in mammalian cells, was found in patients with familial PPB [7]. In the last three decades, we have witnessed the extensive study and confirmation of associations between *DICER1* genetic variants in carriers and the development of a wide range of neoplastic and non-neoplastic conditions. In 2022, the World Health Organization (WHO) described the DICER1 syndrome as an “autosomal dominant tumor predisposition syndrome caused by heterozygous germline pathogenic variants in *DICER1*” [8,9].

The most recent literature on *DICER1* mutations and DICER1-driven thyroid disease in both pediatric and adult thyroid nodules revealed thyroid disease as the most common manifestation of *DICER1* mutations, particularly thyroid follicular nodular disease (TFND), which is now considered most common manifestation of *DICER1* mutations. This finding in a young patient should trigger further evaluations and surveillance. Other manifestations highly suggestive of DICER1-syndrome consist of follicular adenoma with papillary architecture, differentiated thyroid carcinomas (DTC), including follicular thyroid carcinoma and papillary thyroid carcinoma, and thyroid disease with numerous overlapping pathological processes. Two other neoplastic processes, poorly differentiated thyroid carcinomas (PDTC) and thyroblastomas, are specific manifestations of somatic DICER1-related disease [10]. This summary will focus on the *DICER1* gene, its syndromic manifestations, and the distinction between germline/syndromic and somatic DICER1 thyroid manifestations.

## 2. The *DICER1* Gene

The *DICER1* gene is located on chromosome 14q32.13 and is comprised of 1922 amino acids and 27 exons (Figure 1) [11]. DICER1 activity is central to the biogenesis of microRNA (miRNA) which plays a crucial role in the control of protein translation (Figure 2) [12]. This gene encodes a multi-domain endoribonuclease, or dicer protein, including RNAse IIIA and IIIb domains cleaving 3p and 5p miRNAs, respectively, at specific residues E1320, E1564, E1813, and D1709. The 3p and 5p miRNAs are complexed with proteins to become an RNA silencing complex which acts to target messenger RNA and either destabilize their translation or target them for degradation [13,14].

*DICER1* typically presents as either a tumor suppressor gene, resulting from loss-of-function mutations, or as an oncogene, stemming from gain-of-function mutations [13,15]. The majority of germline loss-of-function mutations are inherited, with 10–20% seemingly originating de novo. In individuals with the DICER1 syndrome, most tumors arise in individuals with one inherited *DICER1* mutation with an additional acquired somatic missense *DICER1* mutation within the 5′ “hot-spot” codons in the RNAse IIIb domain (D1705, D1709, D1713, G1809, and E1813), ultimately activating the PI3K/AKT/mTOR pathway [15,16]. The mutational changes impact the 3p’ and 5p’ miRNA ratio, leading to an alteration in mRNA products [11,13,14,17,18]. While the described mutational pattern is the most common in DICER1-associated tumors, mosaicism for missense variants in these same hotspot codons has also been identified, and it is linked to a more severe phenotype [17]. A less common variant of DICER1 syndrome, arising from low-level mosaicism for hotspot RNase IIIb mutations, results in the development of tumors in younger individuals and an elevated occurrence of multisite disease [17,19]. The complete understanding of the penetrance for each of the DICER1-associated neoplasms in inherited conditions is not yet established.

Khan et al. proposed a stepwise model (Figure 3) in which biallelic mutations in *DICER1* lead to an increase prevalence of benign thyroid nodules which over time acquire genetic alterations, prompting a malignant transformation of these nodules [20]. This is supported by nodules of TFND from individuals with DICER1 syndrome showing nodules harboring germline mutation and a second somatic hotspot mutation, with different second hotspot mutations detected in different nodules.

## 3. *DICER1* Gene Alteration-Related Pathology

Heterozygous *DICER1* germline mutation causes DICER1 autosomal dominant familial tumor syndrome. DICER1 syndrome is associated with a wide variety of neoplastic and non-neoplastic conditions. Pleuropulmonary blastoma (PPB) was the initial manifestation linked to DICER1 syndrome, and subsequently, other neoplasms such as pediatric cystic nephroma, ovarian Sertoli–Leydig cell tumor (SLCT), cervix embryonal rhabdomyosarcoma (cERMS), Wilms tumor, embryonal nasal chondromesenchymal hamartoma, ciliary body medulloepithelioma, pituitary blastoma, pineoblastoma, and sarcomas at various sites, including the uterine cervix, kidney, and brain, were associated with the syndrome [21,22]. A better understanding of *DICER1* variants as the cause of DICER1 syndrome and DICER1 syndrome-related endocrine manifestations has also prompted the study of DICER1-driven thyroid disease in both pediatric and adult thyroid nodules. DICER1-related pediatric thyroid disease consists of TFND, follicular adenoma with papillary architecture, DTCs particularly follicular thyroid carcinoma and papillary thyroid carcinoma (macrofollicular type or classical type), PDTC of infancy and childhood, and thyroblastomas [10].

When evaluating a thyroid nodule, the clinicians should pay special attention to any previous history as well as family history of DICER1-related neoplasms. This would increase the suspicion for *DICER1* mutation and prompt proper evaluation. Often, the presence of a *DICER1* mutation is detected by molecular testing at the time of thyroid FNA [23]. While it is crucial to identify individuals with DICER1 syndrome for surveillance purposes, it’s essential to note that most individuals with DICER1 syndrome are either healthy or experience only minor DICER1-associated conditions. Genetic testing can be conducted for at-risk family members to aid in identification and monitoring. Pathologists often play a pivotal role as the first members of the clinical team to encounter familial cancer syndromes during their initial presentation. Therefore, recognizing the histologic features of DICER1-associated disease is becoming increasingly crucial in routine pathology practice for effective patient management and subsequent family testing.

PPB initially sparked interest in studying *DICER1* mutations and their familial associations. PPB is an infrequent multicystic tumor that originates from the peripheral or distal sacs of the lung, characterized by cuboidal cells and a subepithelial layer of small primitive round cells [6,22]. Approximately 70% of patients with PPB have a germline *DICER1* variant [24,25]. *DICER1* mutations may elucidate the cystic stage in PPB, as evidenced by the induction of lung cysts resembling type I PPB in a mouse model following *DICER1* inactivation. [26]. Some studies propose that a biallelic *DICER1* mutation alone may not be adequate for the progression of type I PPB to types II or III. It is suggested that additional mutations, such as *TP53* and *NRAS* mutations, are likely necessary for this progression [19,27].

The genitourinary system is a hotspot of DICER-1 related neoplasms. The most common ovarian tumor in the DICER1 syndrome is the Sertoli–Leydig cell tumor (SLCT) manifesting as a moderately to poorly differentiated adnexal mass with occasional metacystic patterns similar to type I PPB [22,28,29]. Gynandroblastoma, a rare sex-cord tumor with a poorly differentiated Sertoli–Leydig cell pattern and an adult or juvenile granulosa cell tumor pattern, predominantly occurs in the ovary with only rare examples in the testis [30,31,32]. SLCTs also present in association with another neoplasm of the female genital tract, cervical embryonal rhabdomyosarcoma (cERMS) [33]. cERMS present in pubertal or post-pubertal adolescent girls and young women with uterine bleeding and/or a single or multiple botryoid/polypoid mass, which, microscopically, is composed of undifferentiated small round and spindle cells with foci of anaplasia, rhabdomyoblastic, and chondroid differentiation within a myxoedematous stroma [22,34,35]. Nephrogenic neoplasms are another manifestation of DICER1 syndrome. Pediatric cystic nephroma (pCN) is a multiloculated cystic neoplasm presenting at or before 4 years of age as a unilateral, well-demarcated renal mass composed of septate cystic structures with entrapped benign tubular structures similar to type I PPB [36,37]. The stroma is devoid of any immature nephroblastic elements which is the essential distinguishing histologic feature from the cystic partially differentiated nephroblastoma [36,38]. The commonly known Wilms tumor, which usually has an onset before 2 years of age, can also manifest in association with *DICER1* mutations.

Pituitary blastoma is another primitive-type tumor which may coexist with PPB and cystic nephroma. The patients present with features of Cushing disease and an embryonic stage pituitary gland mass [39]. Pineoblastoma is another primitive tumor that is associated with the syndrome.

The gastrointestinal system is commonly affected in several familial syndromes like Cowden syndrome, Lynch syndrome, familial adenomatous polyposis, among others, and DICER1 tumor predisposition syndrome is no exception. Polyps in association with DICER1 syndrome have presented as hamartomatous polyps [22]. The cystic presentation of DICER1 syndrome in the gastrointestinal tract occurs in the liver as a cystic hepatic neoplasm. This multicystic lesion has the architectural and histologic features of type I PPB, although it is surrounded by a cambium layer of rhabdomyoblasts and a concentric fibrous stroma [22,40].

In general, DICER 1-associated neoplasms should guide the clinician and pathologist to suspect either germline or somatic *DICER1* mutations (Table 1). The finding of DICER1 syndrome-associated neoplasms such as PPB, cystic nephroma, SLCT, TFND and DTC in children or adolescents, cERMS, gynandroblastoma, and pituitary blastoma have a high specificity for germline alterations of the *DICER1* gene or DICER1 syndrome. Conversely, neoplasms like TFND and DTC in adults, juvenile intestinal polyps, and Wilms tumor have low specificity for germline *DICER1* alteration.

In the thyroid gland, the presence of two specific neoplasms, thyroblastoma and PDTC of childhood and adolescence suggests a non-syndromic somatic *DICER1* mutation as a genetic driver of the disease.

Some DICER1 syndromic neoplasms also follow a pattern of presentation closely aligned with the patient’s age (Figure 4). Patients with DICER1 syndrome who are less than 10 years old more commonly present with various sarcomas, pineoblastoma, ciliary body medulloepithelioma, meduloepitheliomas, PPB, lung cysts, hamartomatous polyps, and Wilms tumors. Patients with pituitary blastoma are usually diagnosed by the first year of life due to the early Cushing signs. On the other hand, patients over the age of 10 characteristically present with nasal hamartomas or SLCT. Thyroid neoplasms, such as TFND and DTC, cystic nephromas, anaplastic sarcomas and embryonal rhabdomyosarcomas do not follow a particular age pattern.

The overall survival rate of patients with *DICER1* syndrome was 92.9% at a three-year follow-up, with deaths after the three-year period happening on 7% of patients caused by PPB type II, pituitary blastoma, SLCT, rhabdomyosarcoma, and stromal and sex cord ovarian tumors [41].

## 4. Thyroid-Related *DICER1* Gene Alteration Pathology

*DICER1* plays an important role in normal thyroid gland development, and multiple thyroid abnormalities have been identified in DICER1 syndrome [10]. Thyroid nodules are infrequent in children; however, the likelihood of malignancy is higher than in adult thyroid nodules [42].

In 2022, the WHO incorporated a new category for familial thyroid carcinomas into the classification of thyroid neoplasms, prompted by publications identifying these entities [9]. Familial thyroid carcinomas are categorized into C-cell derived familial medullary thyroid carcinomas (MEN2A or 2B and pure familial medullary thyroid carcinoma syndrome) and follicular-cell-derived familial thyroid carcinomas [43]. Clinicopathological correlations have resulted in the further subclassification of the latter into two groups [44]. The first group encompasses a range of familial syndromes distinguished by a prevalence of non-medullary thyroid tumors. This includes conditions like pure familial papillary thyroid carcinoma (PTC) with or without oxyphilia, familial PTC with papillary renal cell carcinoma, and familial PTC with multinodular goiter [44].

Within the second group are syndromes marked by a predominance of non-thyroidal tumors. This includes conditions such as familial adenomatous polyposis, Cowden syndrome, Werner syndrome, Carney complex, and Pendred syndrome. Carney complex, an autosomal dominant disease caused by germline inactivating mutations in *PRKAR1A,* presents as multiple nodules with prominent centripetal hyperplasia and oncocytic cell changes. Cowden/*PTEN* hamartoma tumor syndrome presents as bilateral and multifocal thyroid tumors [45]. McCune–Albright syndrome, which is a germline mosaicism activating mutation of *GNAS*, presents as TFND. Finally, DICER1 syndrome has been added to this group due to the recognition of the involvement of *DICER1* gene alterations in thyroid disease. Germline or somatic *DICER1* mutations each have characteristic thyroid manifestations. Somatic *DICER1* mutations give rise to thyroblastoma and childhood onset PDTC; whereas DICER1 syndrome-related manifestations include TFND, follicular adenoma with papillary architecture, PTC and FTC, pediatric thyroid nodules, as well as childhood onset PDTC.

*DICER1* is now recognized as a driver of pediatric and adult thyroid nodules. Thyroid carcinomas associated with DICER1 syndrome were initially reported only in patients with a previous history of chemotherapy for the treatment of other DICER1 syndrome-related tumors, such as PPB [46]. In 2011, Rio et al. evaluated the association between familial TFND and *DICER1* mutations, concluding that individuals who carry a *DICER1* germline mutation have increased predisposition to develop TFND/MNG, which is among the most highly penetrant phenotypes of the disorder with an associated significantly increased risk of developing thyroid cancer compared to the general population [47]. After Rio’s report, subsequent reports supported their findings [20,48,49]. The newly described follicular adenomas with papillary architecture have been reported to be associated with *DICER1* mutations, and a subset of these were reported in patients with no previous history of chemotherapy for other DICER1 syndrome-related pathologies [46,50,51]. Although far less common, more aggressive tumors such as pediatric poorly differentiated thyroid carcinoma and thyroblastoma have been shown to also harbor DICER1 mutations [46,52,53,54].

The most encountered thyroid lesions associated with *DICER1* gene alterations (Figure 5):

## 5. DICER1 Syndrome Thyroid Related Pathology

### 5.1. Thyroid Follicular Nodular Disease/Multinodular Goiter

The term thyroid follicular nodular disease introduced in the 2022 WHO Classification of Thyroid Neoplasms was created to account for all multifocal/neoplastic lesions that commonly occur in the clinical setting of so-called multinodular goiter. The differential diagnosis in the setting of TFND includes benign and malignant lesions with thyroid carcinoma occurring in 5–15% of the cases. For TFND cases with underlying genetic susceptibility, two specific loci have been identified: one on chromosome 14q (*DICER1* on 14q32.13) and the other on Xp22 [55,56]. In a study conducted by Chong and colleagues, it was discovered that out of 14,993 fine-needle aspirations (FNA) of thyroid nodules, 214 (1.4%) revealed a *DICER1* hotspot mutation. Among these *DICER1* hotspot-positive nodules, an additional pathogenic variant in *DICER1* was identified in 76% of cases. These variants included frameshift, nonsense, missense, and in-frame mutations, as well as loss of heterozygosity. Conversely, the *DICER1* hotspot-negative group showed no other *DICER1* variants upon full *DICER1* sequencing [57]. These findings suggest that *DICER1* alterations are present in a subset of adult thyroid nodules, hinting at the possibility of occult DICER1 syndrome in adults with thyroid nodules. However, as germline analysis was not conducted in this study, the current understanding does not clarify what percentage of the second *DICER1* mutations are germline.

The study by Rio et al. showed an association between *DICER1* and familial TFND [47]. Prospectively, Khan and colleagues evaluated 145 *DICER1* carriers and 135 family controls and found that those with a germline *DICER1* mutation showed a higher cumulative incidence of TFND. The incidence of TFND by the age of 40 was 75% in women and 17% in men; conversely, the incidence in the control group was shown to be 8% in women and 0% in men by age 40 [20]. Despite *DICER1* being classically associated with PPB, the expansion of our knowledge about the predisposition tumor syndrome and its associated tumors has shown that TFND is the syndrome’s most penetrant presentation, especially in females, with a calculated 10–20% penetrance in *DICER1* carriers [58]. Familial TFND as well as TFND in children and adolescents should always prompt suspicion of an underlying germline *DICER1* alteration [59]. Individuals carrying *DICER1* mutations have a 16- to 24-fold increased risk of developing thyroid carcinoma [20,60].

TFND is observed as a common condition even in the absence of *DICER1* alterations. However, specific clinical and morphologic findings can serve as indicators, prompting further molecular investigation. Oliver-Pettit and colleagues conducted *DICER1* testing in a series of eight families referred for childhood-onset TFND or DICER1-related tumors with a familial history of TFND in relatives. In all probands and several of their relatives, germline pathogenic *DICER1* gene variants were identified. Moreover, all tissues studied exhibited clonal pathogenic variants in hotspot regions of *DICER1* [61]. TFND in children and adolescents should consistently raise suspicion of a potential underlying germline *DICER1* alteration.

In the context of a *DICER1* germline mutation, TFND is histologically characterized by the presence of multiple and bilateral nodules exhibiting follicular proliferations. These nodules may manifest as adenomatous nodules, macrofollicular-pattern nodules (Figure 6), well-circumscribed adenomas, and/or nodules displaying intrafollicular centripetal papillary growth. This growth pattern is often referred to as papillary hyperplasia or papillary adenoma, and notably, it lacks the nuclear features typical of papillary thyroid carcinoma (Figure 7) [46]. In patients presenting with numerous adenomatous nodules, Cowden syndrome becomes a pertinent differential diagnosis [62]. However, it’s worth noting that in *DICER1*, TFND nodules typically exhibit a hyperplastic appearance [61]. Suspicion of DICER1-related pathogenesis should be heightened when variable involutional changes are identified in the non-nodular thyroid parenchyma (Figure 8) [46].

### 5.2. Follicular Adenoma with Papillary Architecture

A separate benign entity presenting as a solitary thyroid nodule was described by the WHO in 2022 [8,9,63]. These recently described follicular adenomas with papillary architecture, which have been reported to account for approximately 3% of thyroid nodules, are usually benign cystic follicular cell-derived neoplasms which are non-invasive and encapsulated. They are characterized by intrafollicular, broad, and edematous papillary infoldings with embedded subfollicles (Sanderson’s pollsters), which lack the nuclear features of papillary thyroid carcinoma (i.e., nuclear clearing, peripheral margination of chromatin, intranuclear grooves, and pseudoinclusions). These autonomously hyperfunctioning nodules commonly have peripheral scalloping consistent with rapid resorption of colloid for active hormone synthesis. Due to papillary architecture and atypical nuclear features, some cases are mistaken for papillary thyroid carcinoma.

Multiple etiologies for this specific entity have been described. Approximately 70% of these nodules harbor an activating *TSHR* mutation. Few arise in the context of TFND and some have been associated with *GNAS* alterations in McCune–Albright syndrome, *PRKAR1A* in Carney complex, *PTEN* in Cowden/*PTEN* hamartoma tumor syndrome, and *DICER1* mutations [8,45,64]. In patients with DICER1 syndrome, germline *DICER1* mutations are the most prevalent, although somatic *DICER1* mutations can also occur presenting as multiple and bilateral thyroid nodules.

### 5.3. DICER1-Related Differentiated Thyroid Carcinoma

It was initially thought that radiation therapy was the etiologic mechanism behind pediatric patients with PPB who develop DTC. Now we know that DTC could manifest in the absence of radiotherapy and/or chemotherapy in a *DICER1* carrier. The cumulative model in which biallelic mutations in *DICER1* with superimposed genetic alterations lead to malignant transformation may explain the reason why DTC is also a common finding in DICER1 syndrome [20,60].

Follicular thyroid carcinoma (FTC) and papillary thyroid carcinoma (PTC) in the pediatric population are clinically and genetically distinct from those in adults. In the adult population, *RAS* is the predominant mutation found in FTC and the follicular variant PTC (FVPTC) whereas *DICER1* mutations are found in only 5–8% of follicular adenomas and follicular thyroid carcinomas [65,66]. However, *DICER1* mutations are more prevalent in two conditions: The first is a follicular-patterned thyroid tumor in children (particularly if younger than 10 years old) where a follicular thyroid carcinoma could be the first manifestation of DICER1 syndrome [60,67,68]. Another clue for *DICER1* mutation in the context of FVPTC is coexistent nodular hyperplasia and/or follicular adenoma. These conditions should prompt genetic testing for *DICER1* mutations. The second condition is macrofollicular-predominant follicular thyroid carcinomas in any age group where 75% (six out of eight) of these neoplasms were shown to have *DICER1* mutations along with an additional mutation in most cases [46,51,69]. A study by Onder and colleagues evaluating 56 patients with pediatric PTCs and no clinical or family history of DICER1-related syndromic manifestations showed that all except one DICER1 case were female, 63% of those were FVPTCs, and 37% were classic PTCs [70]. Additionally, no distant metastasis was identified in patients with *DICER1*-altered PTCs.

Familial-syndromic PTC is reported in 10–14% of cases, and in some instances, a germline mutation is identified, as seen in familial adenomatous polyposis, Cowden syndrome, Carney complex, and DICER1 syndrome. In contrast to many non-thyroidal tumors associated with *DICER1* mutations, the pathology of hyperplastic nodules and DTC lacks specific histologic features that distinguish them from their non-*DICER1* mutated counterparts. Mitotic activity may not necessarily correlate with an aggressive clinical course. Consequently, it is crucial to consider the possibility of a *DICER1* germline carrier in cases of so-called nodular hyperplasia/TFND or DTC, especially in patients under the age of 40.

The most encountered thyroid lesions associated with DICER1 syndrome (Figure 9):

## 6. Somatic *DICER1* Alteration Thyroid-Related Pathology

### 6.1. DICER1-Related Poorly Differentiated Thyroid Carcinoma of Infancy and Childhood

Most DICER1-related thyroid findings are of an indolent course. The exception to this rule is PDTC in children and thyroblastoma which are two entities with some overlapping features. The adult-type PDTC is characterized by invasive growth, which includes capsular or vascular invasion, along with solid/trabecular/insular growth. Additionally, it lacks the nuclear features typical of PTC. To be classified as adult-type PDTC, the tumor must exhibit one of the following features: a mitotic count of three or more per ten high power fields (HPF), tumor necrosis, or convoluted nuclei (Figure 10). In adults, *RAS* mutations are the most frequent driver mutations in PDTC. The occurrence of PDTC in young individuals is rare and their clinical and histopathologic features, genetic landscape and outcomes remain largely unknown. Chernock and colleagues studied six PDTCs defined by the Turin criteria in 21-year-old patients for genetic and histologic patterns [54]. All six tumors had solid, insular, or trabecular growth patterns with high mitotic grade and five of six showed tumor necrosis. Pediatric PDTCs lack convoluted nuclei and adult-type molecular alterations. Next generation sequencing identified somatic hotspot mutations in *DICER1* in five of six tumors and whole exome sequencing identified one tumor with a germline pathogenic *DICER1* variant and one with loss of heterozygosity for *DICER1*. Importantly, of these early onset PDTCs, no common mutations characteristic of adult onset PDTC or DTC (*BRAF, RAS, TERT, RET/PTC*, etc.) were detected. These results could indicate that early onset PDTC has a strong association with *DICER1* mutations, and its appearance should lead clinicians to refer patients for genetic counselling.

*DICER1* mutations alone may not necessarily predict a poor outcome in a subset of pediatric PDTCs. The presence of additional genomic alterations may indeed contribute to worse outcomes, as suggested by recent studies indicating that these tumors exhibit invasive growth limited to the thyroid parenchyma but lack vascular invasion [8]. The risk escalation of *DICER1* mutations should integrate the presence of additional genetic events and well-established pathologic variables to ensure predictive dynamic risk stratification in *DICER1*-mutant pediatric PDTCs.

### 6.2. DICER1-Related Thyroblastoma

Thyroblastoma is a recently identified embryonal thyroid neoplasm characterized by highly aggressive biological behavior. This primary primitive thyroid malignancy resembles early fetal embryology and lacks teratoid elements, distinguishing it from what was previously diagnosed as malignant thyroid teratoma or carcinosarcoma [52,53,71,72]. The 2022 WHO definition characterizes thyroblastoma as an embryonal high-grade thyroid neoplasm comprising primitive thyroid-like follicular cells enveloped by a small cell component and a mesenchymal stroma, showcasing variable differentiation [8,9]. In contrast to pleuropulmonary blastoma (PPB), thyroblastoma is not attributed to germline mutations as seen in DICER1 syndrome. Instead, it is more commonly associated with somatic mutations in the *DICER1* gene, with such mutations detected in all examined cases of thyroblastoma.

Ultrasound imaging reveals a prominent, solid nodule that nearly occupies an entire thyroid lobe. Computed tomography (CT) of the neck typically indicates the presence of a sizable nodule extending beyond the thyroid surface or into the substernal area. Macroscopically, the mass typically presents as a sizable, fleshy, soft, solid structure with a red-brown color, effectively replacing the entire lobe. Fine-needle aspiration (FNA) cytology reveals crowded, atypical epithelial cells with a high nuclear to cytoplasmic ratio, arranged in various architectural patterns such as rosette-like microfollicular, solid, and morular patterns. Furthermore, the background contains a small population of atypical mesenchymal cells [23]. Thyroblastoma comprises three distinct cellular components: first, a component represented by fetal-type cells that exhibit positivity for TTF, PAX8, and focal thyroglobulin; second, primitive-appearing thyroid follicles; and third, primitive small round to oval cells organized into irregularly communicating solid aggregates and sheets, featuring areas of necrosis and displaying brisk mitotic activity [73].

The background of thyroblastoma incorporates a primitive spindle cell stroma with variable cellularity, organized into fascicles. This stroma exhibits positivity for SMA and desmin, while a myogenin stain produces equivocal results. Notably, it is negative for TTF1, PAX8, and thyroglobulin (Figure 11). Foci of cartilage are observed in half of thyroblastoma cases; however, distinctive well-differentiated adult-type organoid structures, such as teratomatous components like pilosebaceous elements and skin adnexa, are notably absent.

The most encountered thyroid lesions associated with *DICER1* somatic mutations (Figure 12):

## 7. Conclusions

In summary, as the thyroid manifestations in the context of *DICER1* mutations became recognized, the expanded knowledge of thyroid diseases lead us to better understand its pathogenesis and disorders. Our knowledge of thyroid diseases and *DICER1* gene alterations has led to newly described entities as well as the reclassification of some thyroid diseases in the WHO’s 5th Endocrine edition [9]. The presence of TFND in the context of a pediatric patient should raise the possibility of DICER1 syndrome and lead the clinician to run more genetic evaluations. Additionally, besides the previously well described pathology in multiple organ systems related to DICER1 syndrome, other thyroid findings, like TFND, associated with follicular adenoma with papillary architecture, and DTC, should raise the concern for germline or syndromic DICER1. Finally, PDTC of infancy and childhood and thyroblastoma are two remarkably specific manifestations of somatic DICER1-related disease with poor prognosis.

## Figures and Tables

**Figure 1 cancers-16-00347-f001:**
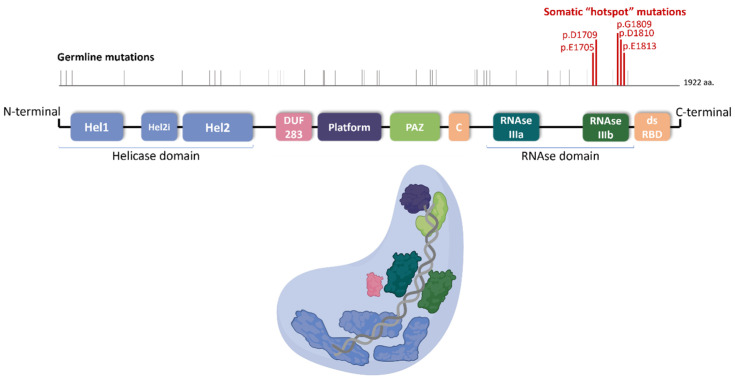
*DICER1* gene including common hotspot mutations.

**Figure 2 cancers-16-00347-f002:**
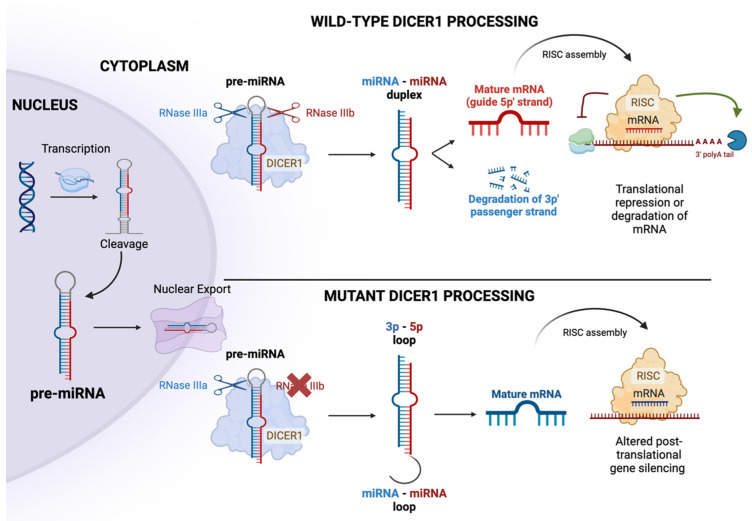
Wildtype and mutant *DICER1* processing mechanisms. The process begins with the transcription of DNA sequences into RNA sequences, forming a precursor miRNA with a characteristic ‘hairpin’ structure. Subsequently, these precursor miRNAs are transported out of the nucleus by exportin 5 and undergo further processing by DICER1 and its accessory proteins. Following this processing, the hairpin structure is degraded, leaving a single, linear piece of miRNA (the complementary piece is degraded within the cell). The resulting single piece is then bound by the RNA-induced silencing complex (RISC). The RISC-miRNA complex, in turn, binds to target mRNA strands, effectively inhibiting translation by the ribosome or degradation of mRNA. A mutated form of DICER1 disrupts the cleavage of the pre-miRNA complex, leading to the formation of a miRNA–miRNA loop. This loop triggers the degradation of the guide 5p’ strand, thereby restricting its assembly with RISC. Consequently, this alteration interferes with post-translational gene silencing.

**Figure 3 cancers-16-00347-f003:**
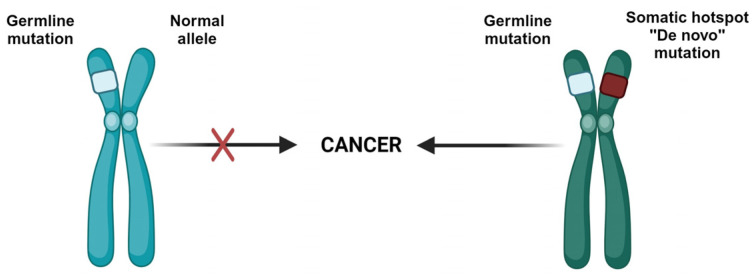
Stepwise transformation model in *DICER1* related TFND.

**Figure 4 cancers-16-00347-f004:**
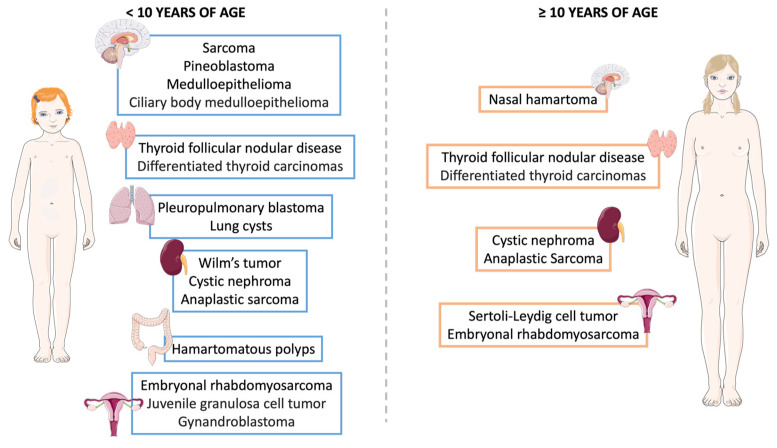
Manifestations of DICER1 syndrome classified by age. In DICER1 syndrome, patients under 10 years old commonly exhibit various sarcomas, pineoblastoma, ciliary body medulloepithelioma, medulloepitheliomas, PPB, lung cysts, hamartomatous polyps, and Wilms tumors. Conversely, patients aged 10 and older typically present with nasal hamartomas or SLCT. Thyroid neoplasms, including TFND and DTC, cystic nephromas, anaplastic sarcomas, and embryonal rhabdomyosarcomas do not exhibit a specific age pattern.

**Figure 5 cancers-16-00347-f005:**
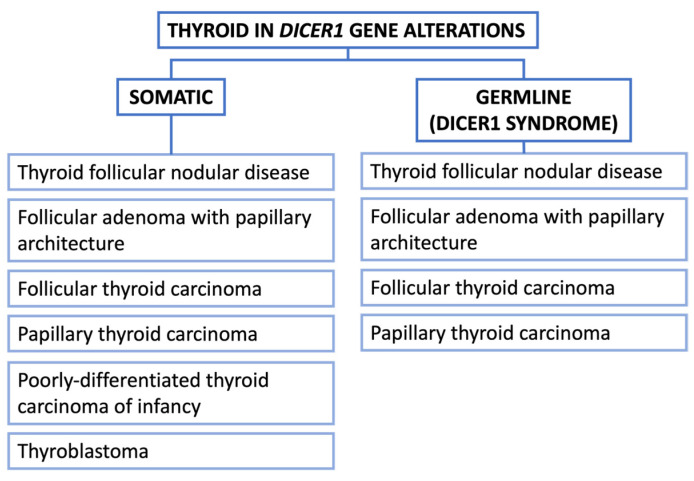
Thyroid manifestations of *DICER1* alterations.

**Figure 6 cancers-16-00347-f006:**
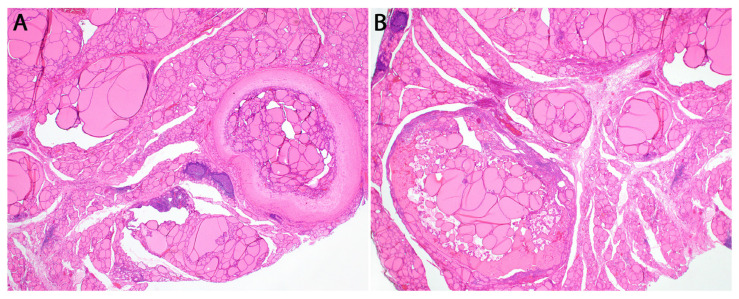
Pictures at different levels of magnification of multinodular thyroid in a patient with a germline *DICER1* mutation (DICER1 syndrome). (**A**) Hematoxylin and eosin (H&E) stained photomicrograph at 20× magnification of well circumscribed hyperplastic nodules with abundant colloid. (**B**) H&E stained photomicrograph at 20× magnification of hyperplastic nodules with variably sized cystic spaces, abundant colloid, and papillary hyperplasia characterized by centripetal growth.

**Figure 7 cancers-16-00347-f007:**
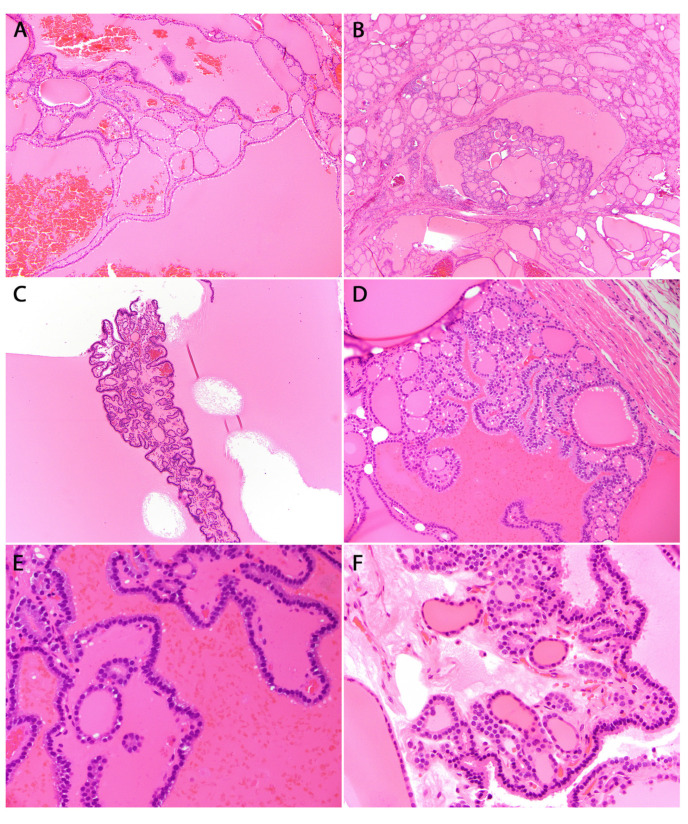
Examples of papillary centripetal growth patterns (**A**–**D**). H&E stained photomicrograph at 100×, 40×, 100×, and 200× magnification, respectively, of a mixture of follicular and papillary architecture with organized centripetal orientation. (**E**,**F**) H&E stained photomicrograph at 400× magnification highlighting the absence of PTC nuclear features.

**Figure 8 cancers-16-00347-f008:**
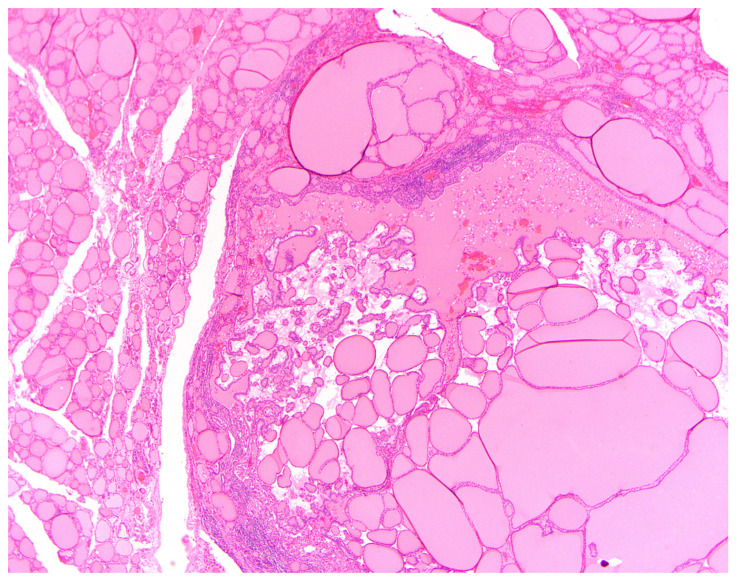
H&E stained photomicrograph at 40× magnification of involutional changes in the non-nodular thyroid parenchyma.

**Figure 9 cancers-16-00347-f009:**
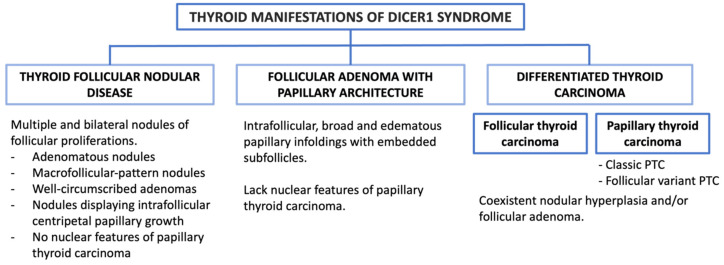
Thyroid manifestations of DICER1 syndrome.

**Figure 10 cancers-16-00347-f010:**
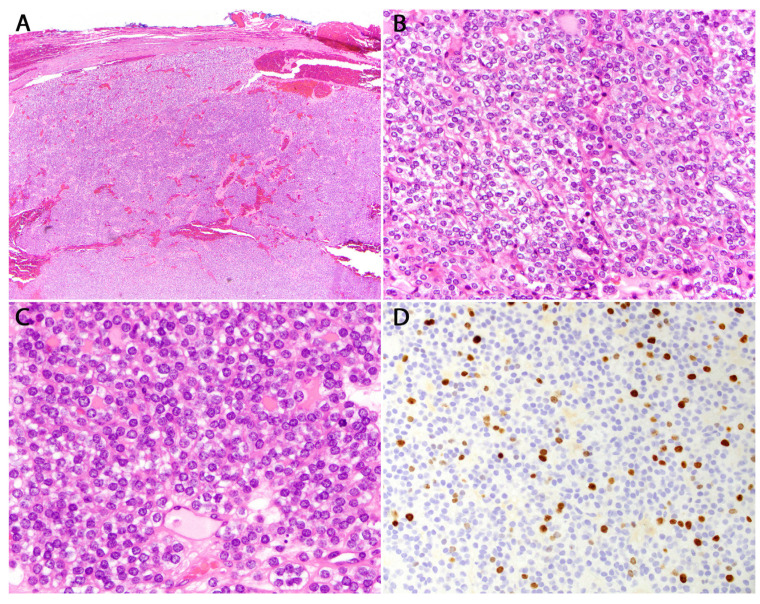
Poorly differentiated thyroid carcinoma of infancy and childhood. (**A**) H&E stained photomicrograph at 40× magnification showing a solid growth pattern. (**B**) H&E stained photomicrograph at 400× magnification showing a trabecular growth pattern. (**C**) H&E stained photomicrograph at 600× magnification highlighting small nuclei lacking PTC nuclear features. (**D**) Ki67 stain at 400× magnification showing a high proliferation rate.

**Figure 11 cancers-16-00347-f011:**
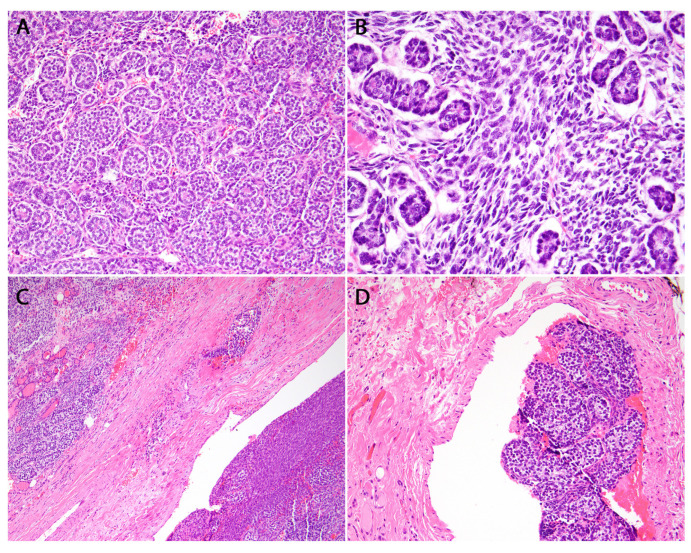
Thyroblastoma. (**A**) H&E stained photomicrograph at 200× magnification showing the embryonal epithelial component characterized by primitive-appearing follicles with limited pink colloid in the lumens. (**B**) H&E stained photomicrograph at 400× magnification of the stromal component characterized by intervening cellular spindle cells. (**C**,**D**) H&E stained photomicrograph at 40× and 100× magnification, respectively, of abundant lymphovascular invasion. (**E**) Positive TTF-1 stain at 200× magnification in the embryonal epithelial component. (**F**) Focally positive Desmin stain at 400× magnification in the primitive stromal component.

**Figure 12 cancers-16-00347-f012:**
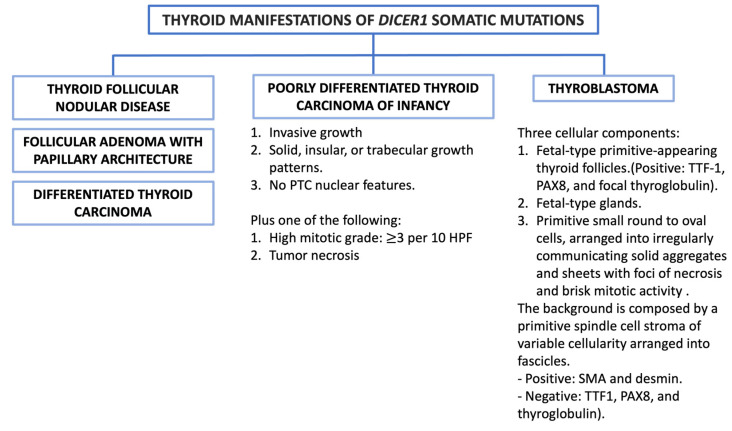
Thyroid manifestations of *DICER1* somatic mutations.

**Table 1 cancers-16-00347-t001:** Summary of the most common DICER1-related tumors and their level of association with DICER1 syndrome.

Associated with Germline Alterations (DICER1 Syndrome)		Associated with Somatic *DICER1* Mutations (Non-Syndromic)
High Specificity	Low Specificity	
Pleuropulmonary blastoma Cystic nephroma Sertoli–Leydig cell tumor **Thyroid follicular nodular disease in children and adolescents (<18 years old)** **Differentiated thyroid carcinoma in children and adolescents (<18 years old)**	Thyroid follicular nodular disease in adults Juvenile intestinal polyps Wilms tumors	Thyroblastoma Poorly differentiated thyroid carcinoma in children and adolescents

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
