# Peer review of "Expanding Our Knowledge of DICER1 Gene Alterations and Their Role in Thyroid Diseases"

_cancers, 2024, doi:10.3390/cancers16020347_

Round 1

Reviewer 1 Report

Comments and Suggestions for Authors

I read with great interest the manuscript. I think it is greatly acceptable in the current form. Please have a look into Cancers - 2023 — DICER1 Syndrome: A Multicenter Surgical Experience and Systematic Review

Author Response

Dear Reviewer#1.

Thank you for reading and evaluating our review paper. 

We found the suggested manuscript very helpful and we added it as a valuable reference to our review. 

Best wishes, 

The authors. 

Reviewer 2 Report

Comments and Suggestions for Authors

The review is well written and covers very well the DIRCR1-related pathologies. I have only minor changes to suggest:

-please use the italics when discussing the gene  and capital letter when referring to human protein.

-figure legends of figure 2 and 4 shoudll be expanded to clarify the models.

- Figures 6-12: please ishow scale bars and magnification used for each picture.

-Check for additional spaces between words ex in line 160 and 204.

- Phrases in lines 231-232 and 412 are incomplete. Please add in the sentence "as shown in picture /table xx." 

Author Response

Dear Reviewer#2.

Thank you for your constructive feedback and valuable suggestions. We appreciate the time and effort you dedicated to reviewing our manuscript. We have carefully considered your comments and made the following revisions to address your concerns:

  1. The original manuscript had the "DICER1" gene italicized, however they were removed at some point of the journal formatting. We proceeded to italicize them again.
  2. We further expanded the descriptions on the models for Fig 2 and 4.
  3. Magnifications were added to the descriptions of all histologic images. 
  4. Additional spaces were removed.
  5. We would appreciate further clarification on this last point. It is the preference of the authors to reference images by including them in a parenthesis. 

We hope these changes adequately address your concerns. Please feel free to review the revised manuscript, and we welcome any further suggestions you may have.

Once again, we sincerely appreciate your insightful comments and look forward to the opportunity to improve our work collaboratively.

The authors. 

Reviewer 3 Report

Comments and Suggestions for Authors

The authors addressed the functional relationship between CD47 and IFT57 in thyroid cancer.

This review systematically summarizes thyroid tumors associated with alterations of DICER1 gene, which is involved in processing of RNA interference (RNAi). DICER1 mutations have been noted to be important mutations in a variety of cancers and are among the key driver genes in thyroid cancer. The article reviews the latest articles and will give readers important and interesting information.

#1. As for somatic mutations of DICER1, examples of hot spot mutations are described, but no examples of mutations are described for germ-line mutations. The authors should show examples of germ-line DICER1 mutations and indicate them in Figure 1.

#2. The lower panel of the Figure 1 seems to indicate the DICER1 protein. Is each domain depicted and colored in the lower panel (3D structure) identical to the domain shown above (2D structure)?

Author Response

Dear Reviewer#4.

Thank you for your constructive feedback and valuable suggestions. We appreciate the time and effort you dedicated to reviewing our manuscript. We have carefully considered your comments and here are our comments:

1. Germline mutations in DICER1 are numerous and span the entire gene. A detailed exploration of these mutations is outside the scope of this review. The primary focus of this figure is to illustrate the general structure of the gene, its 3D conformation, and emphasize the RNAse IIb domain as the most common site for somatic mutations in DICER1.

2. Indeed, the colors in the 3D structure align with those in the 2D domain representation, facilitating easy interpretation for the reader.

Please feel free to review the revised manuscript, and we welcome any further suggestions you may have.

Once again, we sincerely appreciate your insightful comments and look forward to the opportunity to improve our work collaboratively.

The authors. 

Reviewer 4 Report

Comments and Suggestions for Authors

This is an excellen overview about the DICER1 gene alteration. The geneticalk background and explanation is obvious. I have a theoretical question from the clinical side. 

In the postcovid period much more multinodal follicular goiter is removed thank before. Can you see any correlation between the covid infection and a possible DICER1 gene alteration? 

Author Response

Dear Reviewer#4.

Thank you for your constructive feedback and valuable suggestions. We appreciate the time and effort you dedicated to reviewing our manuscript. 

As of our knowledge, there is no literature data post-Covid19 pandemic evaluating DICER1 mutations in the diagnosis of thyroid follicular nodular disease (TFND). With the increasing accessibility of genetic evaluation to the general population, it is anticipated that more DICER1 mutations may be detected in these cases, potentially warranting enhanced surveillance for other malignancies. Presently, there is no routine reflex DICER1 evaluation at the time of diagnosing TFND in patients, therefore a study with this aim is warranted. 

 Please feel free to review the revised manuscript, and we welcome any further suggestions you may have.

Once again, we sincerely appreciate your insightful comments and look forward to the opportunity to improve our work collaboratively.

The authors. 

Reviewer 5 Report

Comments and Suggestions for Authors

The manuscript cancers-2800222 entitled Expanding our knowledge of DICER1 gene alterations and their role in thyroid diseases by Maria Cristina Riascos , is a review about the mutations in DICER1. Recent studies evaluating DICER1 mutation and DICER1-driven thyroid disease in both pediatric and adult thyroid nodules re-vealed thyroid disease as the most common manifestation of DICER1 mutations. Thyroid follic-ular nodular disease and differentiated thyroid carcinomas in infancy are highly specific mani-festations of germline DICER1 mutation or DICER1 syndrome. Furthermore, poorly differentiat-ed thyroid carcinoma and thyroblastoma should raise the concern for somatic DICER1 mutations.

This review work is well written, documented and updated.

Figures and tables are informative.

Minor linguistic revision are recommended.

Line 346, the figure 9 should be centred.

Comments on the Quality of English Language

Minor linguistic revision are recommended.

Author Response

Dear Reviewer#5.

Thank you for your constructive feedback and valuable suggestions. We appreciate the time and effort you dedicated to reviewing our manuscript. 

As recommended, figures were centered and the review will be submitted for english editing. 

Please feel free to review the revised manuscript, and we welcome any further suggestions you may have.

Once again, we sincerely appreciate your insightful comments and look forward to the opportunity to improve our work collaboratively.

The authors.